# Performance Analysis of Short-Span Simply Supported Bridges for Heavy-Haul Railways with A Novel Prefabricated Strengthening Structure

Kaize Xie [1,2,*], Bowen Liu [3], Weiwu Dai [1,2], Shuli Chen [1,2,*] and Xinmin Wang [1,2]

[1] Key Laboratory of Railway Industry of Infrastructure Safety and Emergency Response, Shijiazhuang Tiedao University, Shijiazhuang 050043, China; wangxm@stdu.edu.cn (X.W.)
[2] School of Safety Engineering and Emergency Management, Shijiazhuang Tiedao University, Shijiazhuang 050043, China
[3] School of Civil Engineering, Shijiazhuang Tiedao University, Shijiazhuang 050043, China
[*] Correspondence: kzxie1988@stdu.edu.cn (K.X.); chensl@stdu.edu.cn (S.C.); Tel.: +86-17367918857 (K.X.); +86-13832115905 (S.C.)

**Abstract:** A novel prefabricated strengthening structure (NPSS) is proposed to improve the vertical stiffness and load-bearing capacity of existing short-span bridges for heavier axle-load trains passing through. The strengthening principle of the NPSS is revealed through theoretical derivation. A refined calculation model is prepared to investigate the effects of two important parameters on the structural behavior of the bridge, including the support stiffness and the installation location of the NPSS. The calculation model is also verified with four-point bending test of a bridge removed from a heavy-haul railway. With the calculation model and the response surface methodology (RSM), the functional relationships among the crucial mechanical indexes of the bridge and the two parameters of the NPSS are methodically established. Thus, the optimal values of the parameters are determined via a multi-objective optimization model and the analysis hierarchy process-fuzzy comprehensive evaluation method. Furthermore, the feasibility of the optimal parameters is appropriately verified based on simulations of the vehicle–track–bridge dynamics. The existence of the NPSS with optimal parameters could enhance the vertical stiffness of the bridge by 21.0% and bearing capacity by 19.5%. In addition, it could reduce the midspan dynamic deflection amplitude by 23.4% and vertical vibration acceleration amplitude of the bridge by 25.2%. The results of the study are expected to contribute to the capacity development and rehabilitation of existing heavy-haul railways with low cost and convenient construction without railway outage.

**Keywords:** heavy-haul railway; simply supported bridge; strengthening structure; support stiffness; refined calculation model; multi-objective optimization

## 1. Introduction

As the main passage for transporting bulk goods, such as coal and ore, the heavy-haul railway plays a pivotal role in national economic construction and has become one of the crucial directions of the world's railway development [1]. Heavy axle-load, long-marshaling trains, and high traffic density are the ways to improve transport capacity, and increasing axle load is the most effective measure to reduce costs and improve efficiency [2]. The mechanical performance of existing short-span simply supported bridges for heavy-haul railways is most sensitive to axle load. Therefore, the increase in axle load causes high deflection and crack propagation, which reduces the vertical stiffness and the load-bearing capacity of bridges [3,4]. It also seriously threatens the traffic safety of heavy axle-load trains [5–7]. Meanwhile, the service life of the existing short-span bridges with enormous stock is far from the design value. Considering the economic factor, the existing bridges cannot be demolished or reconstructed. Therefore, bridge-strengthening technologies are

an effective way to improve the performance of the existing short-span bridges to realize the capacity development and rehabilitation of existing heavy-haul railways.

Numerous investigators have examined strengthening theories, experimental methods, materials research and case studies [8–16]. So far, many structure-strengthening approaches have been presented, and can be generally divided into three types, including the section augmentation type, external bonded materials type, and structural system transformation type [17–19]. Some of the methods are also used to improve the seismic performance of the existing structures, and many scholars have carried out substantial research for that reason [19–22]. When the section augmentation method is applied to bridges, it has significant impact on the original structure, train operation and traffic under the bridge [22–25]. Meanwhile, the strengthening effect of the method is suboptimal because of the small change in section size. According to whether prestressing is provided or not, the second type can be further divided into two subtypes, i.e., external non-prestressing method [26,27], and external prestressing method [9,10,28,29]. The bonded materials are mainly steel plates and fiber-reinforced polymer (FRP). Due to its high strength, light weight, noncorrosive nature and good fatigue resistance, the structure-strengthening method with FRP is one of the hottest globe research aspects in the field of structural maintenance [30–32]. Long-term application practice shows that the prestress loss of FRP is relatively large under long-term heavy axle load, which weakens the strengthening effect. In addition, brittle debonding failures and inapplicability on moist surfaces or at low temperatures are also issues that limit the full utilization of this method [9]. Due to its wide adaptability to small-span and long-span bridges, the third type is applied in more and more bridges. Zhou et al. [33], Xu et al. [34], Hu et al. [35] proposed transforming simply supported bridges into continuous systems to improve the transverse stiffness and reduce the deflection of double-T railway bridges or box-girder highway bridges. Song et al. [36] transformed rigid frame-continuous girder into a pseudo-cable-stayed system for mitigating the midspan deflection of the Yellow River Dongming Highway Bridge. Zhang et al. [37] also proposed changing long-span continuous box-girder bridges into self-anchored suspension bridges for controlling both the internal force and deflection of the main girder. Chen et al. [38], Hou et al. [39], and Jiang et al. [40] used external sub-structures to form a new system with the original structure which could substantially upgrade the mechanical behaviors of strengthened bridges. The third type of structure-strengthening approaches offers many desirable properties including high reliability, good fatigue resistance and high environmental adaptability, but the construction period with railway outage is too long to meet the requirement of high traffic density for existing heavy-haul railways.

Thus, a novel prefabricated strengthening structure (NPSS) for short-span simply supported bridges is proposed. The NPSS is constructed to change the support system of bridges by employing an assembly method which has no effect on train operation. Theoretical derivations and finite element simulation are employed to examine the effectiveness of the NPSS. Based on the response surface methodology (RSM) and the multi-objective optimization method, the parameters associated with the NPSS are optimized. Further, the effect of the optimized NPSS on coupled systems, including vehicles, tracks, and bridges, is also investigated on the basis of an inclusive dynamic analysis. The goal of the paper is to determine the key parameters of the NPSS and obtain the optimal values, which can provide guidance for the detailed design of the NPSS. Therefore, the detailed design method of the NPSS is not covered in this paper.

## 2. Proposed Methodology

In order to adapt to the limited installation space and lessen the installation effect on the operation of heavy axle-load trains, an NPSS is proposed for short-span bridges with simply supported ends, as demonstrated in Figure 1.

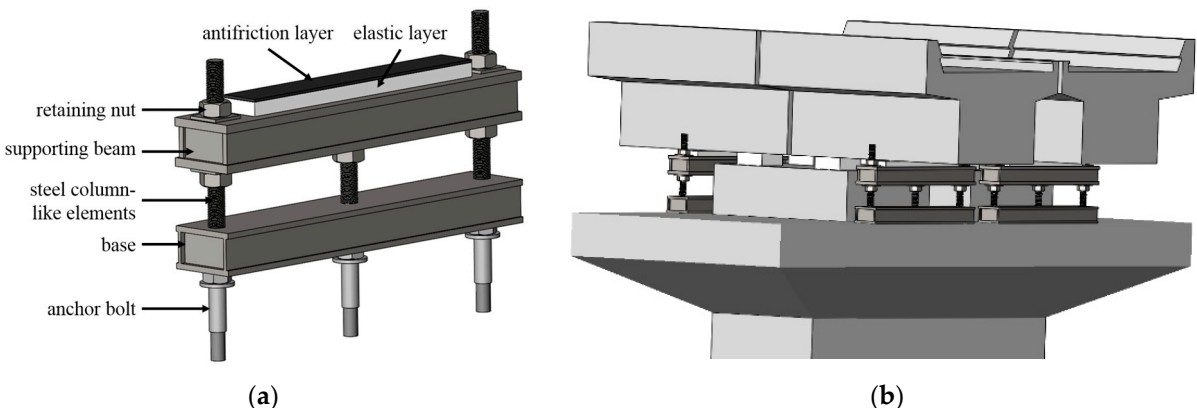

(**a**)                    (**b**)

**Figure 1.** Schematic representation of the proposed NPSS. (**a**) Compositions; (**b**) installation.

The NPSS is essentially composed of a base, steel column-like elements, a supporting beam, and an elastic layer. The base is connected to the pier top by the anchor bolts (see Figure 1a). The steel column-like elements represent an important device that connects the base to the upper supporting beam. The height of the supporting beam could be promptly adjusted with the bolts installed on the steel column-like elements. The elastic layer mounted on the supporting beam could provide an elastically appropriate support for the upper beam. The antifriction layer still should be installed between the elastic layer and the upper beam to lessen the influence of the NPSS on the longitudinal and transverse mechanical performance of the bridge. Since original bridge bearings undergo the dead loads of both the beam and the upper track structure, the NPSS only provides a proper support for the live load of trains. As the structures and sizes of short-span bridges which should be strengthened are dissimilar, the dimensions of the NPSS components should be designed for each bridge. The vertical stiffness and the installation location of the NPSS are the vital factors determining the produced stresses within the NPSS. Additionally, the produced stresses affect the design size of the NPSS, so the two factors denote the main analysis factors.

## 3. Theoretical Model

Based on the load bearing and force transmission paths of the bridge with attached NPSS, the whole system can be simplified into the mechanical model demonstrated in Figure 2. In addition, several simplification hypothesis or assumptions are taken.

(1) The three assumptions including continuity assumption, uniformity assumption and isotropy assumption in mechanics of materials are applied.

(2) The load diffusion caused by rail, fastener, sleeper and ballast bed is ignored.

(3) The support forces of bearing and NPSS are simplified as concentrated loads.

(4) The effect of shear on deformation is ignored.

An appropriate simplified mechanical model can be utilized to show the structural strengthening principle of the NPSS in view of the static performance of the bridge. For short-span bridges, the most unfavorable load condition is that the back bogie of the first vehicle and the front bogie of the second vehicle are symmetrically arranged in the midspan.

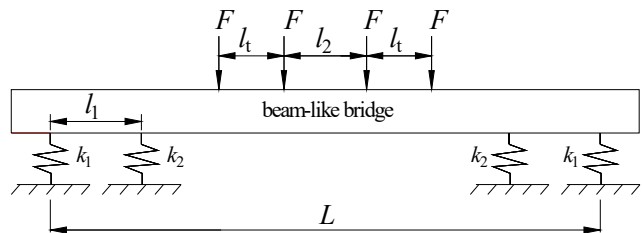

**Figure 2.** Schematic representation of a simplified mechanical model.

Without considering the influence of the shear effect, the midspan bending moment ($M$) and the deflection ($w$) caused by the load accounting for the NPSS are expressed by:

$$M = \left( L - l_t - l_2 - \frac{B}{A}l_1 \right)F, \tag{1}$$

$$w = \left[ C + \frac{B}{A}D \right]\frac{F}{EI}, \tag{2}$$

in which

$$A = \frac{12EI(k_1+k_2)}{k_1k_2l_1} + 6Ll_1 - 8l_1^2$$

$$B = \frac{24EI}{k_1l_1} + 3\left(L^2 - l_2^2\right) - 6l_t(l_t + l_2) - 4l_1^2$$

$$C = \frac{L\left(4l_1^2 - L^2\right)}{24} + \frac{EI(2l_1 - L)}{k_1l_1} + \frac{(2l_t+l_2)^3 + l_2^3}{48}$$

$$D = \frac{EIL(k_1+k_2)}{2k_1k_2l_1} - \frac{EI}{k_1} + \frac{l_1^3}{6} + \frac{L^2l_1}{8} - \frac{Ll_1^2}{3}$$

where $EI$ denotes the vertical bending rigidity of the bridge, $L$ is the computed span of the bridge, as shown in Figure 2, $l_1$ represents the center distance between the bridge bearing and its adjacent NPSS, the factors $l_t$ and $F$ stand for the bogie's wheelbase and the axle load, respectively. $l_2$ is the difference between $l_z$ and $l_t$ (see Figure 2), and $l_z$ denotes the center distance between the back bogie of the first vehicle and the front bogie of the second vehicle. $F$ denotes the wheel weight. Furthermore, the factors $k_1$ and $k_2$ in order are the vertical stiffness values of the bridge bearing and the NPSS (i.e., bearing stiffness and support stiffness for short, respectively). Please see Appendix A for formula derivation.

For large values of the support stiffness, the support force of the bridge bearing and the NPSS subjected to the live load are in opposite directions. Additionally, it causes the NPSS to undergo a large vertical load, which is unfavorable for structural design. Therefore, the support stiffness should guarantee that the support force of the bridge bearing and the resulting force within the NPSS would be in the same direction, namely:

$$k_2 \leq \frac{8EI}{l_1\left[\left(L^2 - l_2^2\right) - 2l_t(l_t + l_2) + 4l_1(l_1 - L)\right]}, \tag{3}$$

Before strengthening, the midspan bending moment ($M_y$) is evaluated by:

$$M_y = (L - l_t - l_2)F, \tag{4}$$

Hence, the reduction of the maximum midspan bending moment ($dM$) caused by the NPSS is obtained as:

$$dM_y = \frac{B}{A}l_1F, \tag{5}$$

The presented relations above denote the main strengthening principle of the NPSS.

## 4. Finite Element Model and Verification

### 4.1. Finite Element Model

The theoretical derivation with the simplified mechanical model fails to consider the influence of some factors such as the track structure, the steel bars in the beam, and the nonlinear characteristics of material and support stiffness. Based on the discrete separated reinforced concrete modeling method, a refined calculation model considering the influence of various factors is established. In view of the symmetry of the structure, only half of the structure is adopted for modeling (see Figure 3).

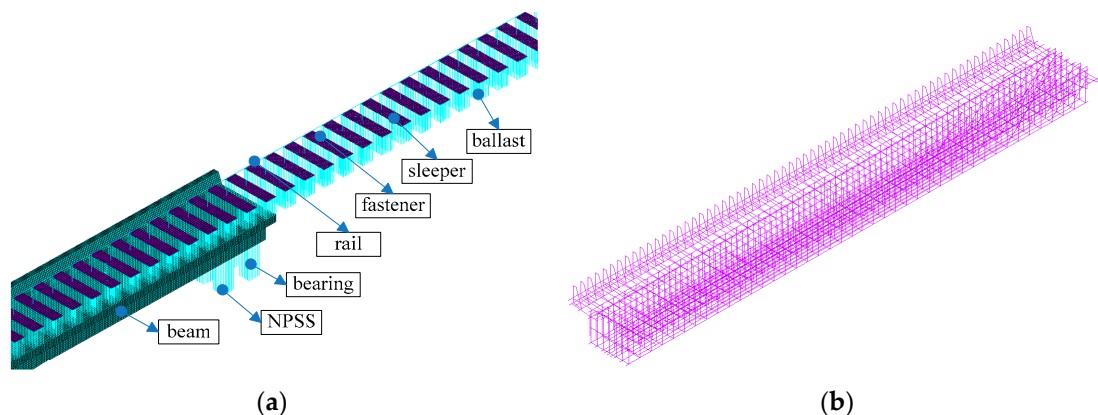

(**a**)                                     (**b**)

**Figure 3.** Refined calculation model. (**a**) Model outline; (**b**) skeleton frame of the steel bars.

Different components in the refined calculation model are simulated with various types of elements according to their mechanical characteristics. The rail is simulated by implementing the Timoshenko beam element, and the torsional freedom is constrained. The sleepers are simulated with shell elements to consider the force dispersion effect. The concrete of the beam is simulated with solid elements, and the stressed and structural steel bars in the beam are simulated by employing the link elements. The connection between the steel bars and the concrete of the beam is appropriately simplified as nodal links. The longitudinal resistance of fasteners, as well as the longitudinal and lateral resistance of ballast bed, could have substantial nonlinear characteristics, so nonlinear spring elements are employed to simulate them. This implies that the NPSS only undergoes pressure force but not tensile force; hence, a nonlinear spring element is also utilized for more rational simulation [41]. The lateral resistance of fasteners, vertical stiffness of fastener, and ballast bed and bridge bearings are simulated with linear spring elements. The nodes connecting the foundation and attaching to the spring elements simulating the bridge bearings, the NPSS and the ballast bed outside the bridge are constrained, which are the boundary conditions of the model. The number of nodes and elements is 220,257 and 269,063, respectively. Table 1 shows statistical information of different elements.

**Table 1.** Information of the model.

| Component | Element Type | Mesh Sizes | Amount |
|---|---|---|---|
| rail | beam188 | 0.01–0.09 m | 2574 |
| sleeper | shell63 | 0.05 m | 16,223 |
| beam | solid185 | 0.05 m | 134,750 |
| steel bars | mesh200/reinf264 | 0.05 m | 55,847 |
| fastener | combin14/combin39 | - | 208/104 |
| ballast bed | combin14/combin39 | - | 39,310/19,655 |
| bearing | combin14/combin39 | - | 196 |
| NPSS | combin39 | - | 196 |

*4.2. Experiments*

In order to verify the feasibility and accuracy of the refined calculation model, a low-height reinforced concrete beam with simply supported ends and a span of 12.5 m is taken as an example. The beam was removed from an existing heavy-haul railway in China, and it has served for more than 20 years. It is a reinforced concrete beam. The cross-section size of the beam is shown in Figure 4. Based on the design data, the concrete of the beam is C30 concrete. The standard values of axial compressive strength and axial tensile strength are 20.1 MPa and 2.01 MPa, respectively. The elastic modulus of the concrete is $3.0 \times 10^4$ MPa. Figure 5 shows the detailing of steel bars.

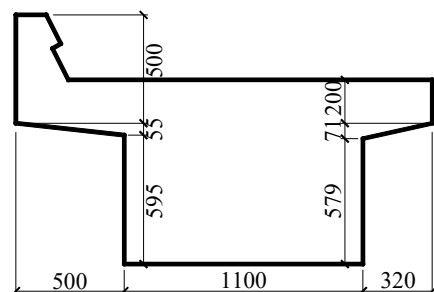

**Figure 4.** Cross-section of the test beam (unit: mm).

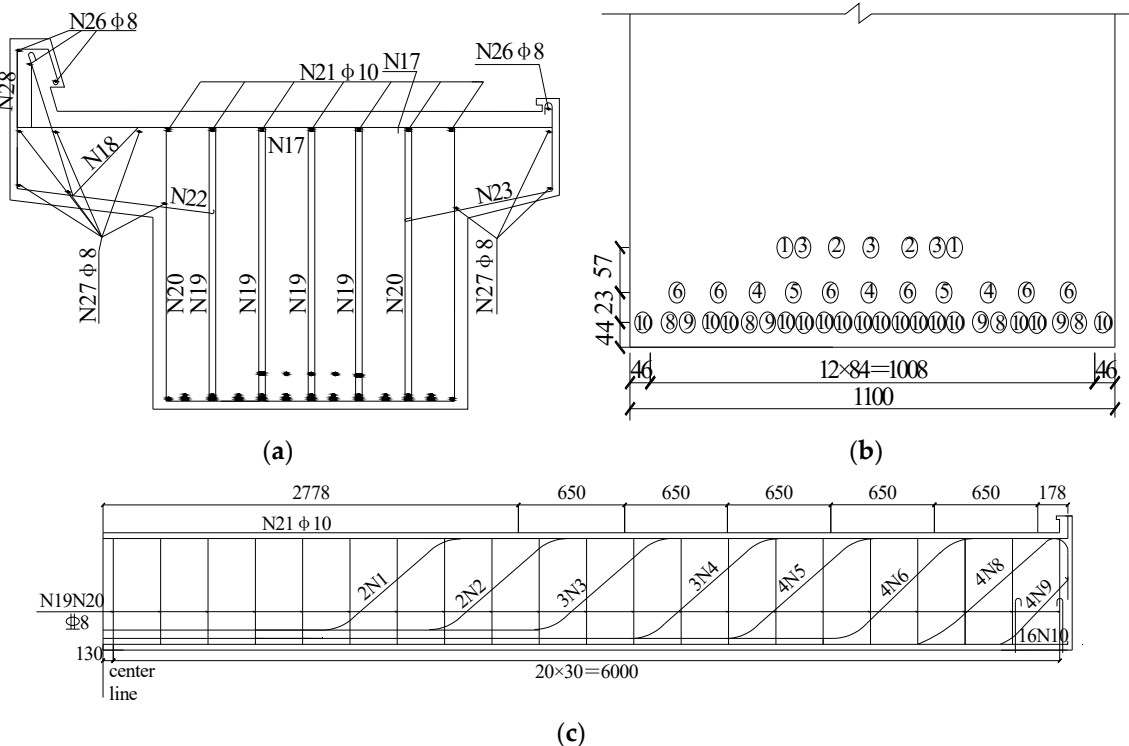

**Figure 5.** Detailing of steel bars. (**a**) Midspan section; (**b**) stressed steel bars of midspan section; (**c**) center half section of beam stem (unit: mm).

Two kinds of steel bars are used in the beam, including hot rolled plain steel bars (HPB) and hot rolled ribbed steel bars (HRB). The yield strengths of the two kinds of steel bars are 300 MPa and 400 MPa, respectively. The steel bars with yield strength of 300 MPa are mainly used as structural reinforcement bars. The diameters of steel bars range from 8 mm to 25 mm. In Figure 6, the principle of the four-point bending test is demonstrated.

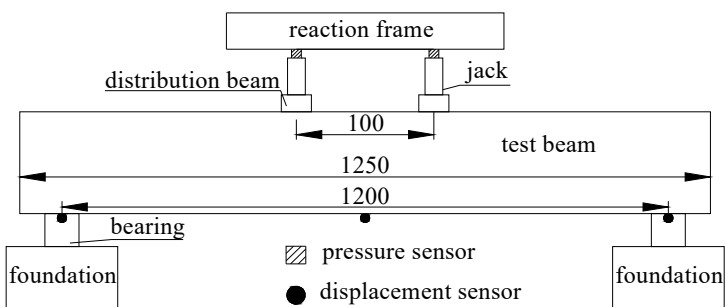

**Figure 6.** Schematic view of four-point bending test (unit: cm).

During the test, four synchronous jacks provide the applied load, and the pressure sensors located between the reaction frame and jacks are also utilized to precisely measure the load. Meanwhile, the displacement sensors located at the cross-section of bearings and midspan are implemented to measure the deflection of various sections of the beam. The detailing of sensors is shown in Figure 6. The parameters of sensors used in the test are shown in Table 2. Figure 7 shows the field test.

**Table 2.** Parameters of sensors.

| Name | Model | Range | Sensitivity | Accuracy |
|---|---|---|---|---|
| pressure sensor | JMZX3420 | 0–2000 kN | 2.0 mV/V | 2 kN |
| displacement sensor (bearings) | CDP-10 | 0–10 mm | 1000 $\mu\varepsilon$/mm | 0.001 mm |
| displacement sensor (midspan) | SDP-50C | 0–50 mm | 100 $\mu\varepsilon$/mm | 0.01 mm |

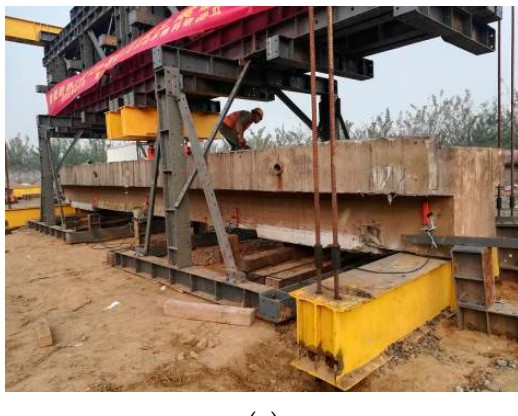 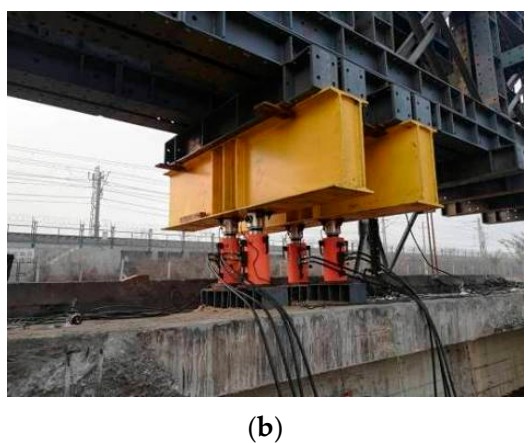

(**a**)        (**b**)

**Figure 7.** Field test. (**a**) Installation of the test beam; (**b**) loading with jacks.

Before the start of the test, preloading is carried out to eliminate the gap between the test beam and equipment. The grading loading method is adopted, and the load increment of each stage is about 10 percent of the estimated ultimate bearing capacity. Each level of load is held for 5 min, and then the test data of sensors is recorded.

*4.3. Comparison between Model and Experimental Data*

Through combining the design parameters of the bridge and the experimentally reported data on material properties, the calculation parameters of the model are summarized in Table 3.

**Table 3.** Calculation parameters of the model under investigation.

| Type | Parameter | Value | Type | Parameter | Value |
|---|---|---|---|---|---|
| steel bars | Young's modulus (Pa) | $2.1 \times 10^{11}$ | bridge section | area (m$^2$) | 1.18 |
| | Poisson's ratio | 0.3 | | vertical moment of inertia (m$^4$) | 0.087 |
| | yield strength (MPa) | 400/300 | single bearing | length × width × height (m) | $0.3 \times 0.3 \times 0.05$ |
| | tangent modulus (Pa) | $1.6 \times 10^9$ | | Young's modulus (Pa) | $6.84 \times 10^8$ |
| | diameter (mm) | 8–25 | | vertical stiffness (N/m) | $1.23 \times 10^9$ |

The plastic characteristics of the concrete and the steel reinforcement of the beam are appropriately taken into account in the structural modeling. The relationships between the stress and strain of concrete under the action of uniaxial loads are illustrated in Figure 8a,b. The Drucker–Prager criterion is employed to model the yielding mechanism of the concrete.

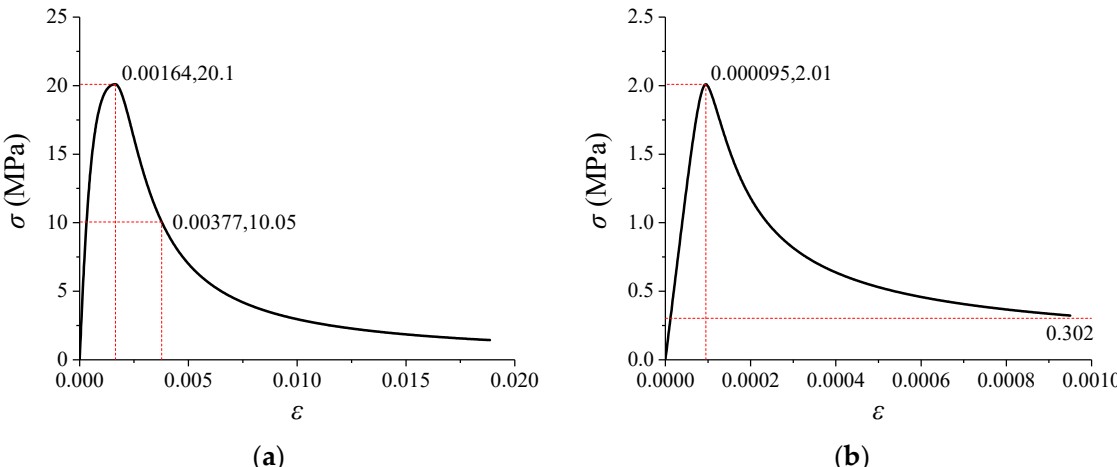

**Figure 8.** Uniaxial stress–strain relationship of the concrete. (**a**) Compression; (**b**) tension.

Figure 9 illustrates the midspan displacement-load curves obtained by the test and the refined model. In order to remain consistent with the test state, the influence of the track structure on the bridge has been ignored in the calculation.

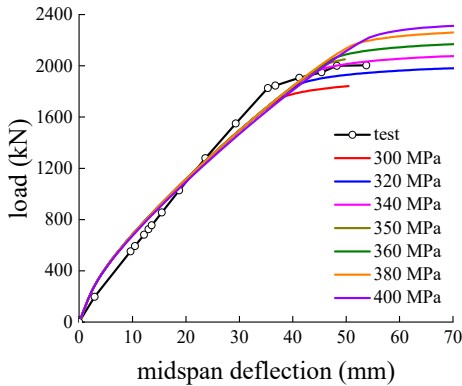

**Figure 9.** The achieved results from the test and the calculation model.

As can be seen from Figure 9, the change rule and the variation trend of the deflection-load curves of both the test and the calculation model are matched, which indicates that the calculation model could be employed for analyzing the interactions between the bridge and the corresponding upper track structure. There exists only a discrepancy at the yield position, essentially resulting from three types of errors. Firstly, the properties of materials at various locations of the test beam are not the same. Secondly, the nonlinear trend of the stress-strain curve and the failure criteria of materials employed in the calculation model still demonstrate an apparent distinction from the actual. Thirdly, the measure error is objective and unavoidable. The effect of yield strength of the steel bars on the inelastic response is most pronounced. Figure 9 shows the deflection-load curves with different yield strengths of the steel bars. In order to determine the appropriate yield strength, Formula (6) is introduced to get the error ($\Delta$) between calculation and test results.

$$\Delta = \frac{\sqrt{\sum\limits_{i=1}^{n} (F_{Ci} - F_{Ti})^2}}{n}, \tag{6}$$

where $F_{Ci}$, $F_{Ti}$ stand for the $i$-th calculation and test data with the same deflection. $n$ denotes the number of comparison data. Figure 10 shows the relation between the error and yield strength of the steel bars.

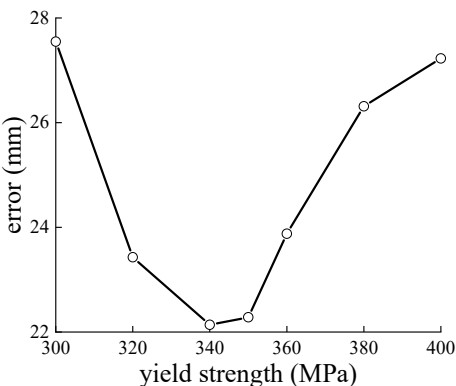

**Figure 10.** Relation between the error and yield strength.

According to the results in Figure 10, the error is smallest when the yield strength is near 340 MPa which is close to the standard value of tensile strength of steel bars. Therefore, this value is used in subsequent analyzes. Considering the failure mode of the test beam and sufficient shear capacity based on the allowable stress method, the existing short-span simply supported bridges will not experience oblique section failure. Thus, shear strengthening measures are not covered in this paper.

## 5. Discussion

### 5.1. Comparison Studies with and without NPSS

A C96 vehicle with an axle load of 30 t is chosen as the loading system. The train loads and the corresponding parameters are illustrated in Figure 11 with more detail. In the calculations, two vehicles pass through the beam regardless of the dynamic effect that simulates various positions of the load relative to the base beam. The value of $l_1$ is set equal to 0.8 m, and support stiffness $k_2$ is set as 0.5 times of the bearing stiffness ($k_1$) which is $2.46 \times 10^9$ N/m. The calculation parameters of the track structure are also achievable in Refs. [42,43].

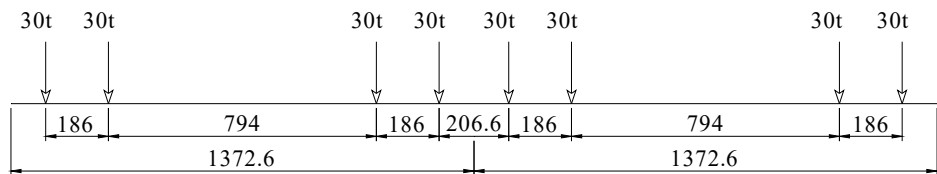

**Figure 11.** Parameters of the applied load (unit: cm).

Figure 11 demonstrates the comparison results of the mechanical performance with and without NPSS. Figure 12a–c represents the midspan deflection, midspan bending moment, and the support force of the bridge bearing and the NPSS, respectively. The abscissa axis of Figure 11 represents the position of the first wheel, and the midspan of the bridge is zero.

According to the plotted results in Figure 12, the midspan deflection and bending moment in the case of using the NPSS are substantially reduced compared to the case without using NPSS. In the cases of with and without NPSS, the maximum midspan deflections in order are 2.9 mm and 6.2 mm, and the associated maximum bending moments are 804.8 kN·m and 1210.8 kN·m, respectively. These results clearly illustrate that the use of the NPSS leads to the reduction of the maximum deflection and bending moments of the midspan by 53.2% and 33.5%, respectively. As shown in Figure 13, the plastic deformation occurs without NPSS, resulting in an irrecoverable midspan deflection of 1.5 mm after the train leaves the bridge. In the case of applying the NPSS, the strengthening beam is mostly in the elastic state due to the decreased bending moment. This is also one of the reasons why the existing bridges should be strengthened during travel of heavy-haul trains. The

directions of the support force of the bridge bearing with and without NPSS are dissimilar. The amplitude of the support force of the NPSS reveals an increase of 116.6 kN (i.e., 28.0%) compared with that of the bridge bearing without NPSS.

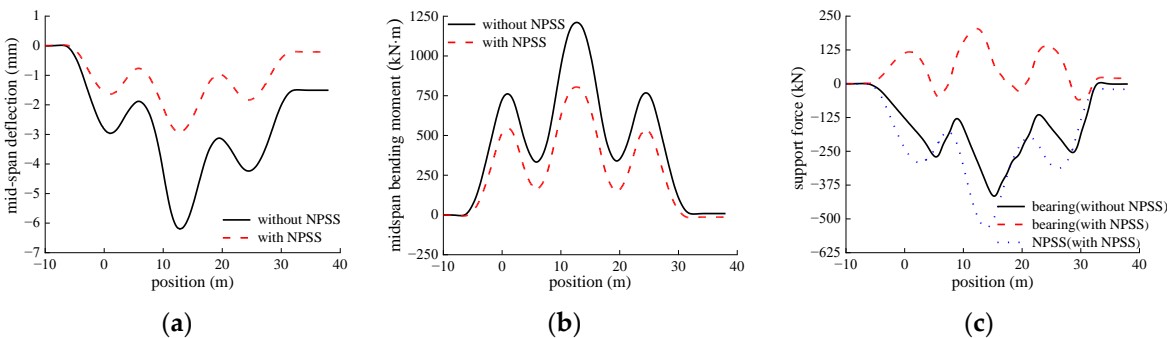

**Figure 12.** The calculated results with and without NPSS. (**a**) Midspan deflection; (**b**) midspan bending moment; (**c**) support force.

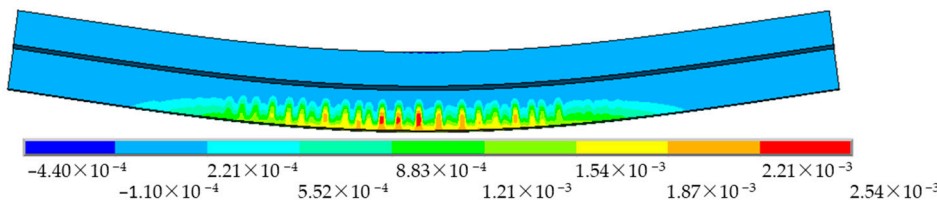

**Figure 13.** Contour plots of the plastic strain of the beam.

In order to make the support forces of the bridge bearing and NPSS in the same direction when the bridge is subjected to the live load, the ratio $k_2/k_1$ should be less than 0.15 by virtue of Equation (3). The excessive support force of the NPSS not only intensifies the difficulty in the structural design and the possibility of pier concrete collapse, but also leads to partial function loss of the bridge bearing. Therefore, it is necessary to optimize the support stiffness to ensure that the NPSS does not undergo too much load while improving the bridge stiffness and bearing capacity.

### 5.2. Influence of the Support Stiffness

In the current investigation, the values of $l_1$ and $k_1$ in order are kept fixed at 0.8 m and $2.46 \times 10^9$ N/m, and five levels are taken for the ratio $k_2/k_1$ (i.e., 0.1, 0.2, 0.3, 0.4, and 0.5). The results by the calculation model are illustrated in Figure 14. Figure 14a displays the support force amplitudes of both the bridge bearing and the NPSS as a function of the ratio $k_2/k_1$. The support pressure force of the bridge bearing is viewed as positive, otherwise negative. Figure 14b is the relational curve between the midspan bending moment and deflection amplitude and $k_2/k_1$.

According to Figure 14a, with the rise of the support stiffness, the support force amplitude of the NPSS and the negative support force amplitude of the bridge bearing grow, while the positive support force amplitude of the bridge bearing lessens. For the ratio $k_2/k_1$ in the range of 0.1–0.5, the support force amplitude of the NPSS increases by 302.9 kN, indicating a growth of 132.0%. For the case of $k_2/k_1 = 0.1$, the support force amplitude of the NPSS and the positive support force amplitude of the bridge bearing in order are 229.5 kN and 212.1 kN. These values are approximately equal, representing that the NPSS and the bridge bearing collaboratively undergo the train load. Additionally, the negative support force amplitude of the bridge bearing is only 30.1 kN, revealing that the whole structure is in good stress performance.

According to Figure 14b, both the midspan deflection and bending moment amplitudes would lessen with the growth of the support stiffness; however, the reduction rate gradually reduces. On the one hand, the increase of the support stiffness leads to enhancement of

the beam's vertical stiffness. On the other hand, it provides a higher support force of the NPSS and the negative support force of the bridge bearing, reducing the bending moment of each section as illustrated in the envelope graph of the bending moment of Figure 15. Because of the reasons explained above, the midspan deflection lessens with the increase of support stiffness. For the ratio $k_2/k_1$ in the interval of 0.1–0.3, the midspan deflection and bending moment amplitudes decrease by 1.2 mm and 154.2 kN·m, respectively (i.e., a drop of 26.0% and 14.9%). However, as $k_2/k_1$ increases from 0.3 to 0.5, the midspan deflection and bending moment amplitudes only exhibit a reduction of 0.5 mm and 73.5 kN·m (i.e., fall by 15.2% and 8.4%). In the case of $k_2/k_1 = 0.1$, the midspan deflection amplitude is obtained as 4.7 mm, which does not exceed the usual value of 4.9 mm under the action of design load [44].

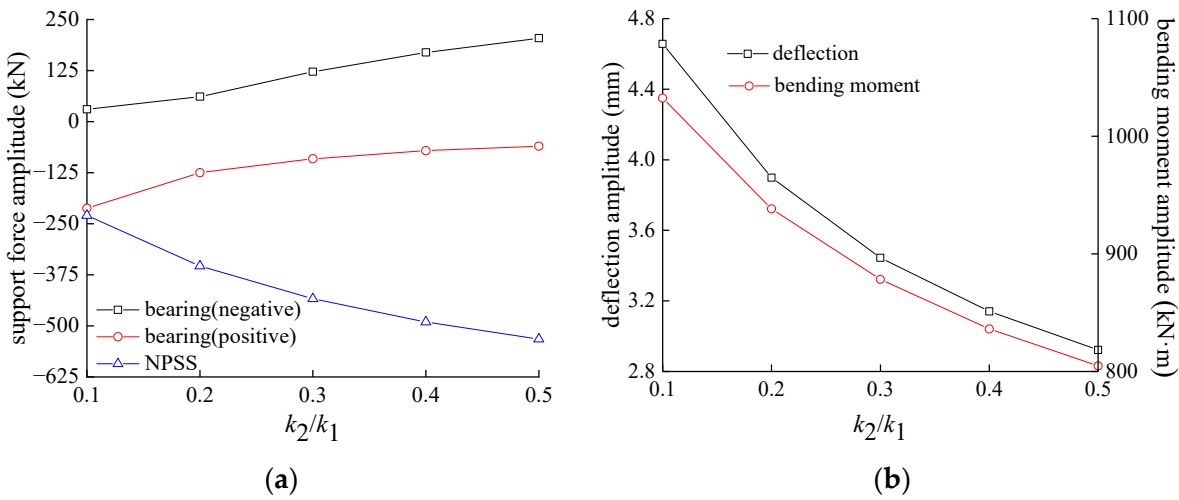

(**a**)　　　　　　　　　　　　　　　　　　(**b**)

**Figure 14.** Effects of the support stiffness. (**a**) Support force; (**b**) deflection and bending moment amplitudes.

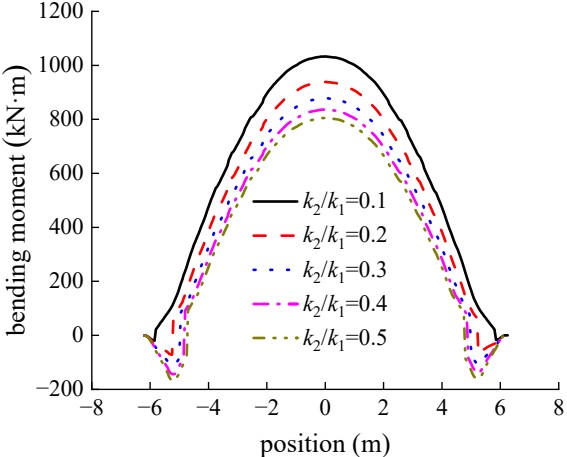

**Figure 15.** Envelope graph of the bending moment for various levels of $k_2/k_1$.

### 5.3. Influence of the Installation Location

Due to the limitation of the size of the pier top, the study range of $l_1$ is almost small, from 0.4 m to 0.8 m. Under the calculation conditions, the ratio $k_2/k_1$ is kept fixed at 0.1. The obtained results have been demonstrated in Figure 16.

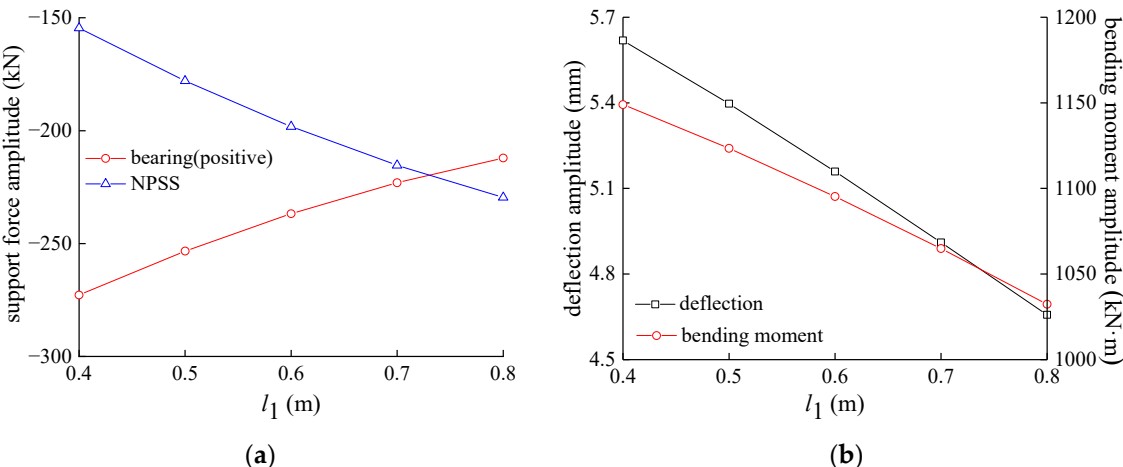

**Figure 16.** Effects of the installation location. (**a**) Support force; (**b**) deflection and bending moment amplitudes.

As shown in Figure 16a, with the growth of $l_1$, the supporting force amplitudes of the bridge bearing and the NPSS vary approximately linearly, where the positive supporting force amplitude of the bridge bearing lessens and the supporting force amplitude of the NPSS grows. For the case of $l_1 = 0.73$ m, the support forces of the bridge bearing and the NPSS are approximately equal. As the value of $l_1$ ranges from 0.4 m to 0.8 m, the support forces of the bridge bearing and the NPSS demonstrate the changes of 60.7 kN and 75.0 kN, respectively (i.e., the percentage alterations of 22.2% and 48.5%).

The illustrated results in Figure 16b reveal that the midspan deflection and bending moment amplitudes lessen linearly with the increase of $l_1$ such that the reduction rates in order are 2.4 mm/m and 291.8 kN·m/m. As the value of $l_1$ increases from 0.4 m to 0.8 m, the midspan deflection and bending moment amplitudes lessen by 1.0 mm and 116.6 kN·m, respectively, signifying a reduction of 17.1% and 10.1%.

Thus, the installation location of the NPSS plays a vital role in the strengthening effect; nevertheless, in the case of $k_2/k_1 = 0.1$, $l_1$ must be greater than 0.7 m to meet the requirements of the bridge deflection.

### 5.4. Parameters Optimization

Although the increase of $k_2$ and $l_1$ is beneficial in reducing both the deflection and bending moment of the beam, it results in a higher support force of the NPSS and a smaller positive support force of the bridge bearing, which is unfavorable to the NPSS. Therefore, the amplitudes of the positive support force of the bridge bearing and the support force of the NPSS should be equal as far as possible. Meanwhile, the midspan deflection and the negative support force of the bridge bearing should be as small as possible to determine the optimal values of $k_2$ and $l_1$. By virtue of the refined calculation model and the numerical test, the RSM is implemented to determine the explicit functional relationship between the mechanical indexes of the bridge with NPSS and support stiffness, and the installation location of NPSS. Thereby,

$$w = 7.406 - 4.410x - 3.438l_1 + 5.331x^2 + 1.162l_1{}^2 - 4.138xl_1 \tag{7}$$

$$F_{\text{b}} = 28.929 + 932.019x - 61.521l_1 - 1196.027x^2 - 262.240l_1{}^2 + 1297.991xl_1, \tag{8}$$

$$F_{\text{y}} = -113.757 + 91.352x + 407.353l_1 + 46.588x^2 - 360.671l_1{}^2 + 433.894xl_1, \tag{9}$$

where $x = k_2/k_1$, $w$ represents the midspan deflection amplitude, $F_{\text{b}}$ denotes the absolute value of the difference between the positive support force amplitude of the bridge bearing

and that of the NPSS, and $F_y$ is the negative support force amplitude of the bridge bearing. The multiple correlation coefficient, modified multiple correlation coefficient, and $R^2$ (prediction) of the fitting results are all over 0.95. By taking the minimum of $w$, $F_b$, and $F_y$ as the objective functions, a multi-objective optimization model could be constructed as:

$$
\begin{aligned}
\text{Min} \ \ & Y = \left[ w(x, l_1), \ F_b(x, l_1), \ F_y(x, l_1) \right] \\
s.t. \ \ & 0 \le w \le 4.9 \\
& 0 \le F_b, F_y \\
& 0.1 \le x \le 0.5 \\
& 0.4 \le l_1 \le 0.8
\end{aligned}
\tag{10}
$$

The Pareto optimal boundary of the multi-objective optimization model is determined with the multi-objective evolutionary algorithm NSGA-II, as illustrated in Figure 17a. Figure 17b presents the optimized values of the variables $x$ and $l_1$.

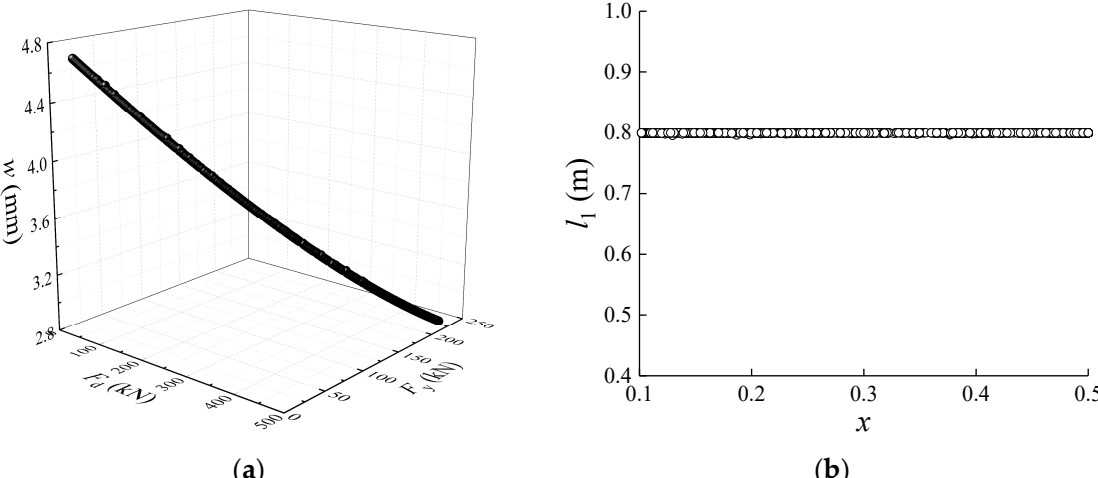

**(a)**          **(b)**

**Figure 17.** The plotted results of the multi-objective optimization. (**a**) Pareto optimal solution; (**b**) optimization variable value.

As shown in Figure 17, the Pareto optimal boundary is a spatial curve. Regardless of the value of $x$, the corresponding value of $l_1$ for the optimal design is fixed at 0.8 m, which is the maximum allowable value. For the optimal boundary given in Figure 16a, the analysis hierarchy process-fuzzy comprehensive evaluation (AHP-FCE) method is employed to determine the reasonable support stiffness of the NPSS for engineering applications. Since $w$, $F_b$, and $F_y$ are all negative indicators, the data standardization scheme shown in Equation (11) is adopted. Take midspan deflection amplitude $w$ as an example:

$$
r_i = 1 - \frac{w_i}{\max(w)} + \left\{ 1 - \max \left[ 1 - \frac{w_i}{\max(w)} \right] \right\},
\tag{11}
$$

where $w_i$ and $r_i$ represent the values before and after data standardization, respectively.

The weight coefficients of $w$, $F_b$, and $F_y$ determined based on the AHP-FEC are 0.15, 0.48, and 0.37, respectively. The weight coefficients remain unchanged with the variation of the selected optimal boundary points. At this time, the values of $x$ and $l_1$ associated with the optimal scheme are 0.1018 and 0.8, which means that the support stiffness value is $2.5 \times 10^8$ N/m, and the center distance between the bridge bearing and its adjacent NPSS is 0.8 m. According to the optimal factors, the midspan deflection-load curve of the bridge based on the four-point bending test has been presented in Figure 18. The plot of the load-bearing capacity of the strengthened bridge demonstrates an increase of 19.5%. The slope of the curve in the elastic deformation stage of the bridge with the NPSS also rises

from 104.7 kN/mm to 126.7 kN/mm, which reveals that the vertical stiffness of the bridge has been enhanced by 21.0%.

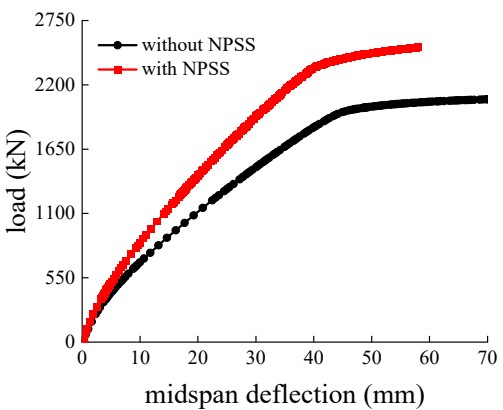

**Figure 18.** The midspan deflection-load curves for the cases of with and without NPSS.

*5.5. Dynamic Performance Verification*

Since the NPSS chiefly influences the vertical mechanical behavior of the strengthened bridge, the vehicle–track–bridge vertical coupling dynamical model with ANSYS/LS-DYAN software is established to prove the validity of the proposed optimization scheme. The dynamics model illustrated in Figure 19 consists of a vehicle subsystem, a tracking subsystem, a bridge subsystem, and a wheel–rail contact system.

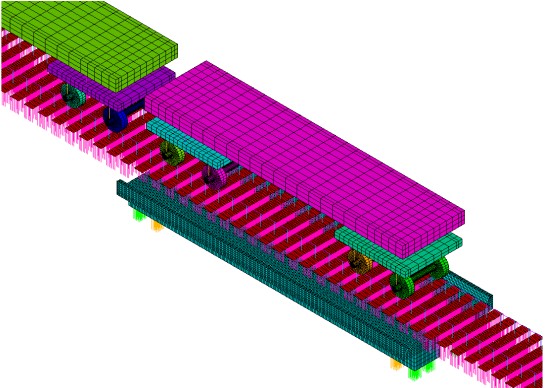

**Figure 19.** A dynamical finite-element-based model for vehicle–track–bridge system.

As the span of the simply supported bridge is smaller than the length of a vehicle, the vehicle subsystem is employed to simulate three-car trains. In order to simplify the calculation, the nonlinear materials' behaviors are ignored. The track vertical irregularity excitation is taken in accordance with the American 5th-grade track spectrum and the running speed is set as 60 km/h. Figure 20 demonstrates the dynamic response of the bridge when the vehicles pass across the bridge.

According to the plotted results in Figure 20, the NPSS has a trivial influence on the trend of dynamic deformation and vibration of the bridge. However, the NPSS can effectively control the midspan dynamic deflection and vibration acceleration amplitude. Through employing the NPSS, the midspan deflection and vibration acceleration amplitudes lessen by 1.16 mm and 0.41 m/s$^2$, which indicates a reduction of 23.4% and 25.2%, respectively. Due to the improvement of the vertical stiffness of the bridge with NPSS, the car's vertical vibration acceleration and wheel unloading rate intensify by 1.31 m/s$^2$ and 0.01, which still satisfies the demand for heavy-haul train running safety.

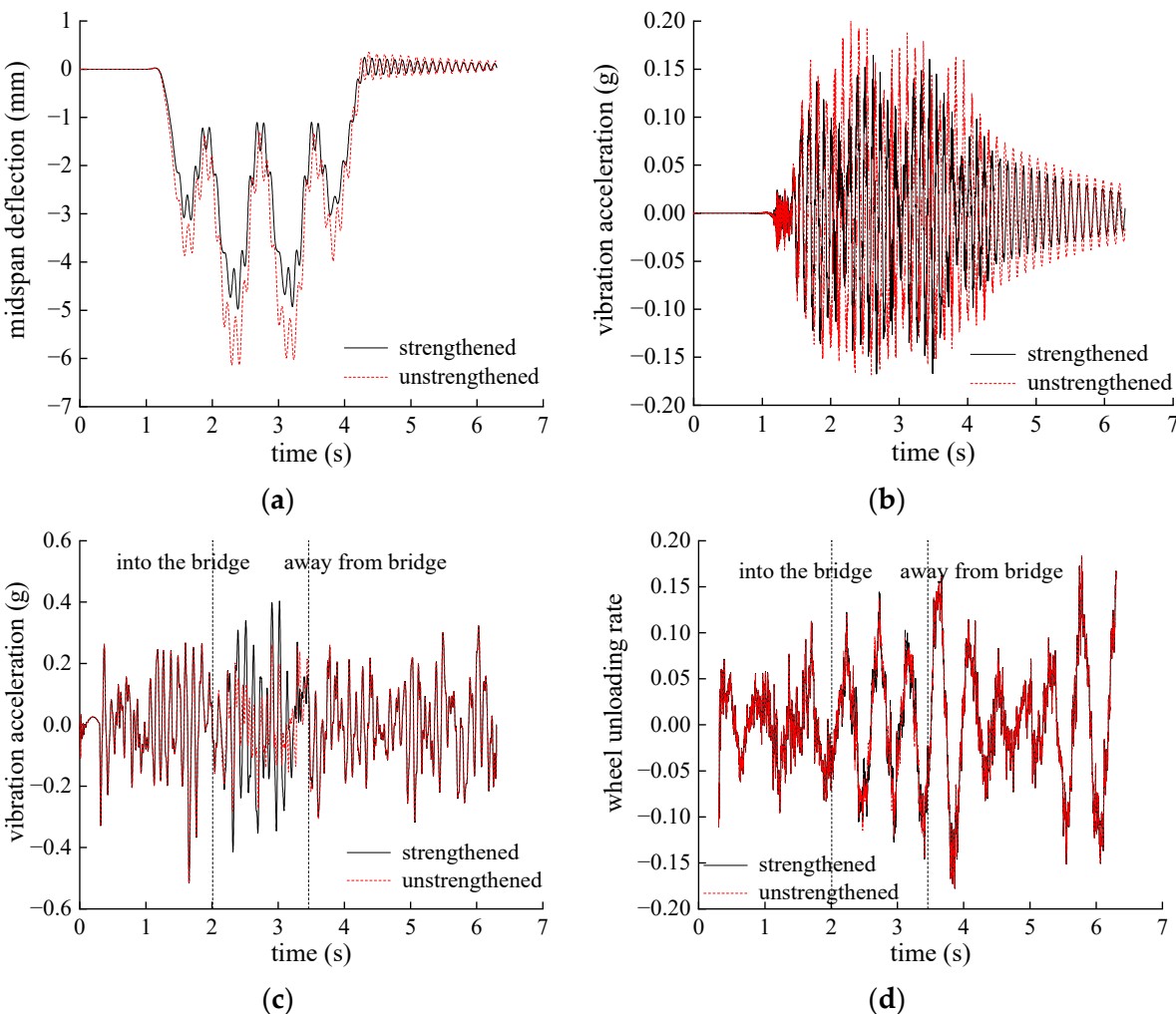

**Figure 20.** Dynamic responses. (**a**) Midspan deflection; (**b**) midspan's vibration acceleration; (**c**) car's vibration acceleration; (**d**) wheel unloading rate.

## 6. Conclusions

A new bridge strengthening structure, called NPSS, which can be constructed using an assembly construction method, has been proposed. Both theoretical derivation and numerical simulation have proven that the NPSS can improve the vertical stiffness and load-bearing capacity of existing short-span bridges for heavier axle-load trains. The effect of two key parameters on structural behavior of the bridge has been investigated, and the parameters were also optimized based on a multi-objective optimization model. The major achieved results are summarized as follows:

(1) The excessive support stiffness of the NPSS would cause the support forces of bearing and the NPSS to reverse, which enables the NPSS to bear additional load. The strengthening effect was not significant when the support stiffness of the NPSS was too small. The support stiffness of the NPSS should be controlled within a reasonable range.

(2) The increase in the center distance between the bridge bearing and its adjacent NPSS would result in a more significant strengthening effect. Therefore, if conditions permit, this distance should be increased as much as possible.

(3) The optimal values of the support stiffness and center distance between the bridge bearing and its adjacent NPSS in order are $1.25 \times 10^8$ N/m and 0.8 m, which could enhance the load-bearing capacity and the vertical stiffness of the bridge by 19.5% and 21.0%, respectively, and reduce the midspan dynamic deflection amplitude and vertical vibration acceleration amplitude of the bridge by 23.4% and 25.2%, respectively.

Although the strengthening method with the NPSS is verified with the static and dynamic calculation models, field test verification is also necessary. Therefore, the NPSS will be produced and installed for a short-span bridge for heavy-haul railway, and field testing will be carried out to verify the feasibility and effectiveness of the strengthening method with the NPSS.

**Author Contributions:** Conceptualization, K.X., S.C. and X.W.; Data curation, B.L. and W.D.; Formal analysis, K.X. and B.L.; Funding acquisition, K.X. and S.C.; Investigation, K.X., B.L. and W.D.; Methodology, K.X. and S.C.; Software, K.X. and X.W.; Writing—original draft, K.X.; Writing—review and editing, B.L., S.C. and X.W. All authors have read and agreed to the published version of the manuscript.

**Funding:** This research was funded by National Natural Science Foundation of China, grant numbers 52008272 and 51978423. This research was also funded by Natural Science Foundation of Hebei Province, grant numbers E2022210046 and E2021210099.

**Data Availability Statement:** All data and models used during the study appear in the submitted manuscript.

**Conflicts of Interest:** The authors declare no conflict of interest.

**Appendix A**

Figure A1 shows the calculating diagram based on Figure 2.

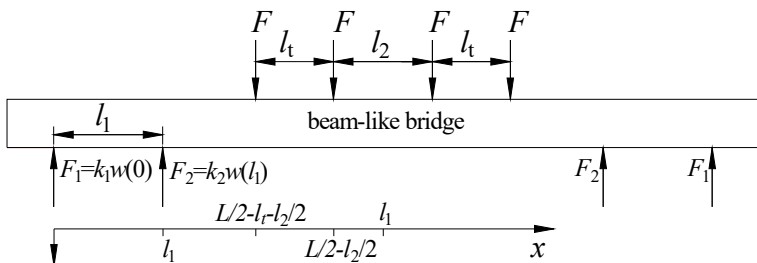

**Figure A1.** Calculating diagram.

In Figure A1, $F_1$ and $F_2$ represent the support force of the bridge bearing and the NPSS. The bending moments at different positions are expressed by:

$$M(x) = \begin{cases} F_1 x & 0 \le x \le l_1 \\ F_1 x + F_2 (x - l_1) & l_1 < x \le \frac{L}{2} - l_t - \frac{l_2}{2} \\ F_1 x + F_2 (x - l_1) - F\left(x - \frac{L}{2} + l_t + \frac{l_2}{2}\right) & \frac{L}{2} - l_t - \frac{l_2}{2} < x \le \frac{L}{2} - \frac{l_2}{2} \\ F_1 x + F_2 (x - l_1) - F\left(x - \frac{L}{2} + l_t + \frac{l_2}{2}\right) - F\left(x - \frac{L}{2} + \frac{l_2}{2}\right) & \frac{L}{2} - \frac{l_2}{2} < x \le \frac{L}{2} \end{cases} \quad (A1)$$

Based on the relation between the second derivative of deflection and the bending moment, the first derivative of deflection and the deflection can be obtained.

$x \in [0, l_1]$

$$\begin{aligned} EIw'(x) &= -\frac{F_1}{2} x^2 + C_1 \\ EIw(x) &= -\frac{F_1}{6} x^3 + C_1 x + C_2 \end{aligned} , \quad (A2)$$

$x \in \left(l_1, \frac{L}{2} - l_t - \frac{l_2}{2}\right]$

$$\begin{aligned} EIw'(x) &= -\frac{F_1}{2} x^2 - \frac{F_2}{2} (x - l_1)^2 + C_3 \\ EIw(x) &= -\frac{F_1}{6} x^3 - \frac{F_2}{6} (x - l_1)^3 + C_3 x + C_4, \end{aligned} , \quad (A3)$$

$$x \in \left( \frac{L}{2} - l_t - \frac{l_2}{2}, \frac{L}{2} - \frac{l_2}{2} \right]$$

$$EIw'(x) = -\frac{F_1}{2}x^2 - \frac{F_2}{2}(x - l_1)^2 + \frac{F}{2}\left(x - \frac{L}{2} + l_t + \frac{l_2}{2}\right)^2 + C_5$$

$$EIw(x) = -\frac{F_1}{6}x^3 - \frac{F_2}{6}(x - l_1)^3 + \frac{F}{6}\left(x - \frac{L}{2} + l_t + \frac{l_2}{2}\right)^3 + C_5 x + C_6 \qquad \text{(A4)}$$

$$x \in \left( \frac{L}{2} - \frac{l_2}{2}, \frac{L}{2} \right]$$

$$EIw'(x) = -\frac{F_1}{2}x^2 - \frac{F_2}{2}(x - l_1)^2 + \frac{F}{2}\left(x - \frac{L}{2} + l_t + \frac{l_2}{2}\right)^2 + \frac{F}{2}\left(x - \frac{L}{2} + \frac{l_2}{2}\right)^2 + C_7$$

$$EIw(x) = -\frac{F_1}{6}x^3 - \frac{F_2}{6}(x - l_1)^3 + \frac{F}{6}\left(x - \frac{L}{2} + l_t + \frac{l_2}{2}\right)^3 + \frac{F}{6}\left(x - \frac{L}{2} + \frac{l_2}{2}\right)^3 + C_7 x + C_8 \qquad \text{(A5)}$$

where $C_1$–$C_8$ are undetermined constants. The curve of the first derivative of deflection and the deflection are continuous at $l_1$, $L/2 - l_t - l_2/2$ and $L/2 - l_2/2$. The relation among $C_1 \sim C_8$ can be determined.

$$\begin{aligned} C_1 &= C_3 = C_5 = C_7 \\ C_2 &= C_4 = C_6 = C_8 \end{aligned} \qquad \text{(A6)}$$

Because of the symmetry, the first derivative of deflection is zero for the case of $x = L/2$.

$$-\frac{F_1}{2}\left(\frac{L}{2}\right)^2 - \frac{F_2}{2}\left(\frac{L}{2} - l_1\right)^2 + \frac{F}{2}\left(l_t + \frac{l_2}{2}\right)^2 + \frac{F}{2}\left(\frac{l_2}{2}\right)^2 + C_7 = 0$$

The above formula can be simplified as:

$$C_7 = \frac{F_1}{8}L^2 + \frac{F_2}{8}(L - 2l_1)^2 - \frac{F}{2}\left(l_t^2 + \frac{l_2^2}{2} + l_2 l_t\right), \qquad \text{(A7)}$$

With the relation among $F_1$, $F_2$ and $w(x)$ and the principle of force balance, Formulas (A8)–(A10) can be obtained.

$$F_1 = k_1 w(0) \Rightarrow F_1 = \frac{k_1}{EI}C_2, \qquad \text{(A8)}$$

$$F_2 = k_2 w(l_1) \Rightarrow \frac{EIF_2}{k_2} = -\frac{F_1}{6}l_1^3 + C_1 l_1 + C_2, \qquad \text{(A9)}$$

$$F_1 + F_2 = 2F \qquad \text{(A10)}$$

Substituting Formulas (A6)–(A8) and Formula (A10) into Formula (A9), Formula (A11) can be obtained.

$$\begin{aligned} \frac{EIF_2}{k_2} = &-\frac{2F - F_2}{6}l_1^3 + \frac{2F - F_2}{8}L^2 l_1 + \frac{F_2 l_1}{8}\left(L^2 + 4l_1^2 - 4Ll_1\right) \\ &-\frac{Fl_1}{2}\left(l_t^2 + \frac{l_2^2}{2} + l_2 l_t\right) + \frac{2FEI}{k_1} - \frac{F_2 EI}{k_1} \end{aligned} \qquad \text{(A11)}$$

Formula (A11) can be simplified as:

$$\left[\frac{EI(k_1 + k_2)}{k_1 k_2 l_1} - 8l_1^2 + 6Ll_1\right]F_2 = \left[\frac{24EI}{k_1 l_1} + 3\left(L^2 - l_2^2\right) - 6l_t(l_t + l_2) - 4l_1^2\right]F \qquad \text{(A12)}$$

Assume that

$$\begin{aligned} A &= \frac{12EI(k_1 + k_2)}{k_1 k_2 l_1} + 6Ll_1 - 8l_1^2 \\ B &= \frac{24EI}{k_1 l_1} + 3\left(L^2 - l_2^2\right) - 6l_t(l_t + l_2) - 4l_1^2 \end{aligned}$$

then $F_1$, $F_2$ and $M(L/2)$ can be obtained as:

$$F_2 = \frac{B}{A}F, F_1 = \left(2 - \frac{B}{A}\right)F,$$

(A13)

$$M\left(\frac{L}{2}\right) = F_1\frac{L}{2} + F_2\left(\frac{L}{2} - 1\right) - F\left(l_t + \frac{l_2}{2}\right) - F\frac{l_2}{2}$$
$$= F\left(L - l_t - l_2 - \frac{B}{A}l_1\right)$$

(A14)

Substituting Formulas (A8) and (A9) into Formula (A2), Formula (A15) can be obtained.

$$EIw\left(\frac{L}{2}\right) = -\frac{F_1}{48}L^3 - \frac{F_2}{48}(L - 2l_1)^3 + \frac{F}{6}\left(l_t + \frac{l_2}{2}\right)^3 + \frac{F}{48}l_2{}^3$$
$$+ \frac{EIF_2L}{2k_2l_1} + \frac{(2F-F_2)Ll_1{}^2}{12} - \frac{(2F-F_2)EIL}{2k_1l_1} + \frac{(2F-F_2)EI}{k_1}$$

(A15)

Substituting Formula (A10), Formula (A13) and Formula (A15) into Formula (A5), Formula (A16) can be obtained.

$$EIw\left(\frac{L}{2}\right) = -\frac{F_1}{48}L^3 - \frac{F_2}{48}(L - 2l_1)^3 + \frac{F}{6}\left(l_t + \frac{l_2}{2}\right)^3 + \frac{F}{48}l_2{}^3$$
$$+ \frac{EIF_2L}{2k_2l_1} + \frac{(2F-F_2)Ll_1{}^2}{12} - \frac{(2F-F_2)EIL}{2k_1l_1} + \frac{(2F-F_2)EI}{k_1}$$

(A16)

Formula (A16) can be simplified as:

$$w\left(\frac{L}{2}\right) = \left[\frac{EIL(k_1+k_2)}{2k_1k_2l_1} - \frac{EI}{k_1} + \frac{l_1^3}{6} + \frac{L^2l_1}{8} - \frac{Ll_1^2}{3}\right]F_2$$
$$+ \left[\frac{L\left(4l_1^2-L^2\right)}{24} + \frac{EI(2l_1-L)}{k_1l_1} + \frac{(2l_t+l_2)^3+l_2^3}{48}\right]F$$

(A17)

Assume that

$$C = \frac{L\left(4l_1^2-L^2\right)}{24} + \frac{EI(2l_1-L)}{k_1l_1} + \frac{(2l_t+l_2)^3+l_2^3}{48}$$
$$D = \frac{EIL(k_1+k_2)}{2k_1k_2l_1} - \frac{EI}{k_1} + \frac{l_1^3}{6} + \frac{L^2l_1}{8} - \frac{Ll_1^2}{3}$$
,

then $w(L/2)$ can be obtained as:

$$w\left(\frac{L}{2}\right) = DF_2 + CF = D\frac{B}{A}F + CF = \left(C + \frac{B}{A}D\right)\frac{F}{EI}$$

(A18)

In order to guarantee that the support force of the bridge bearing and the resulting force within the NPSS would be in the same direction, $F_1$ should be greater than zero, namely:

$$F_1 = \left(2 - \frac{B}{A}\right)F \geq 0 \Rightarrow B \leq 2A$$

(A19)

Substituting $A$ and $B$ into Formula (A19), Formula (A20) can be obtained.

$$\frac{24EI}{k_1l_1} + 3\left(L^2 - l_2^2\right) - 6l_t(l_t + l_2) - 4l_1^2 \leq \frac{24EI(k_1+k_2)}{k_1k_2l_1} + 12Ll_1 - 16l_1^2$$
$$k_2 \leq \frac{8EI}{l_1\left[\left(L^2-l_2^2\right)-2l_t(l_t+l_2)+4l_1(l_1-L)\right]}$$

(A20)

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
