# Peer review of "Performance Analysis of Short-Span Simply Supported Bridges for Heavy-Haul Railways with A Novel Prefabricated Strengthening Structure"

_buildings, doi:10.3390/buildings13040876_

Round 1

Reviewer 1 Report

The reviewer thanks the authors for their contribution. The contents of the article are interesting and useful information. Nevertheless, the article is NOT well formulated and organized and, in the reviewer’s opinion, the goal of the work must be better explained within Abstract, Introduction and Conclusions. Moreover, the publication in the “Buildings, MDPI” is not recommended unless the following suggestions are taken into account:

1) Please, polish the abstract. Please, add sentences to explain the meaning, the main points, the improvement and the promising application of the study. The logic should also be improved.

2) Introduction. The current state of knowledge relating to the topic has not been covered and clearly presented, and the authors’ contributions and findings are not emphasized. In this regard, the authors should make their effort to address these issues.

3) Introduction. In the literature, stiffness of concrete bridge girders has rigorously been studied through dynamic laboratory tests and analytical simulations on several prestressed specimens having different shapes and geometries of tendons. Please, refer to these issues through the following references:

-  https://doi.org/10.1016/j.istruc.2021.10.093

-  https://doi.org/10.1016/j.jsv.2015.11.047

4) Objectives and findings should be presented more clearly (e.g., using the following division of the sections: Introduction, Proposed Methodology, Theoretical Model, Finite Element Model, Experiments, Comparison between Model and Experimental Data, Discussion, Conclusions). The current main sections appear NOT very well organized and divided. Please, review the corresponding parts.

5) The geometric and mechanical characteristics of the prestressed concrete girders have not clearly been illustrated. Please, provide more information and introduce figures with schemes of such structures.

6) Please, report the layout of the experiments with locations of the measuring points by the conventional sensors. Please, specify.

7) The authors used conventional sensors. Please, specify the corresponding characteristics of the measurements, e.g., frequency (or the period) of the recording data etc. Moreover, range, sensitivity, resolution and accuracy of the conventional sensors should be underlined.

8) Please, cite the finite element (FE) software ANSYS in the references.

9) The FE analyses performed by ANSYS must be better explained with, particularly, details of the models used. It is not clear how the model of the prestressed concrete girders are composed (beam elements, plate and shell, etc., with the corresponding amounts and mesh sizes) and how the external loading were applied. Which are the geometric characteristics of such girders ? Which are their boundary conditions ? Moreover, have geometric nonlinear analyses been performed ? How much are the values of the prestress forces applied along the cables ? Please, revise these parts and provide more information about the FE models.

10) Please, insert a table which lists the types of FE used, with the corresponding amounts within the models and mesh sizes.

11) Please, substitute the term “mid-span” with “midspan” within the text.

12) The further work should be mentioned at the end of the article. Please, modify.

Reviewer 2 Report

In this paper, a novel prefabricated strengthening structure (NPSS) for short-span simply supported bridges is proposed. The theoretical derivations and the finite element simulation are employed to examine the effectiveness of the NPSS. Based on the response surface methodology (RSM) and the multi-objective optimization method, the parameters associated with the NPSS are optimized. Further, the effect of the optimized NPSS on coupled systems including vehicles, tracks, and bridges is also investigated on the basis of an inclusive dynamical analysis. In general, the topic is interesting, and a few questions are required to be well handled.

1. The language checking is suggested by the reviewer. For example, Line 14, ‘…do not have insufficient vertical stiffness…’ maybe it is ‘do not have sufficient…’?

2. The authors performed the survey of structural system transformation strengthening method, which is a great work. The reviewer suggests to focus more on the short comings of the existing references. The following papers related to the system transformation strengthening can be included into the Intro. 10.1016/j.jobe.2022.104904; 10.1016/j.soildyn.2010.06.001; 10.1016/j.strusafe.2023.102330;

3. In the 2.2. A Simplified Mechanical Model, what are the simplification hypothesis or assumptions in the theroretical calculations, maybe these can be further enhanced.

4. In the test, how did the authors design the specimens? As it is a common sense that the strengthened specimen will show better than the un-strengthened one, thus the design approach may be a more interesting point to conclude.

Reviewer 3 Report

The study concerns the proposal of a strengthening system for short-span railway bridges. The system is based on introducing new supports on the substructure. The authors perform an extensive numerical analysis to analyse the benefits in terms of mid-span flexure related to the use of the presented components. The study presents several issues preventing the publication.

·        The strengthening strategy is based on the use of additional supports which reduces the clear length of the bridge span. This strategy clearly fosters a reduction of the mid-span moment. However, the study lacks in considering shear effects (evaluated for the axle loads approaching the support) which are significant in short-span beams. The variation of shear at the support should be inevitably considered when using such a retrofit strategy. Appropriate extensive analyses are required.

·        A parametric analysis is carried out to identify retrofit parameters (distances between old and new supports, stiffness of the bearings). However, the parametric analysis does not involve the consideration of constructive and geometric characteristics of the devices presented in Sec. 2. For example, how the device should be designed to achieve the desired stiffness features? Which are the implications of designing the proposed device for a given value of support force to be absorbed? For example, the instability response of the so-called “steel column-like elements” should be checked. Furthermore, the installation methods of the proposed device should be commented on to sustain the feasibility of the approach.

·        The calibration of the numerical model can be improved. It seems the elastic branch is sufficiently calibrated, while the inelastic response of the numerical model occurs for moment values different with respect to the experiment.

·        Formulations at 2.2. should be checked. What is F in Equation 1a? If F is the single axial load the formulation should be revised. How A,B,C, D are calculated?

·        The state-of-the-art is incomplete. Studies on strengthening bridges from all over the world should be included (e.g. 10.1016/j.engstruct.2022.114356, 10.1016/j.engstruct.2022.114827) discussing the advantages and disadvantages of different strategies. Additionally, studies on typical structural issues affecting railway bridges should be commented such as fatigue (10.1016/j.engfailanal.2021.105996) and degradation (10.1080/15732479.2021.1956550).

Round 2

Reviewer 1 Report

The authors carried out the required revisions.

Author Response

Thank you for the feedback provided by the reviewers. We also appreciate your approval of the changes we made. We will follow the suggestions you originally provided for our next research step.

Reviewer 2 Report

The paper can be accepted in the current form.

Author Response

Thank you for the feedback provided by the reviewer. We also appreciate your approval of the changes we made and our research achievements. We will follow the suggestions you originally provided for our next research step. We look forward to exploring more relevant results and having more opportunities to communicate with you.

Reviewer 3 Report

This reviewer suggests including appropriate clarifications in the manuscript on points 1 and 2 of the previous review report.

Author Response

Thank you for the feedback provided by the reviewer. According to the advice of the reviewer, we have appropriate clarifications in the manuscript on points 1 and 2 of the previous review report. The revised content is as follows:

The goal of the paper is to get the key parameters of the NPSS and obtain the optimal values, which can provide guidance for the detailed design of the NPSS. Therefore, the detailed design method of the NPSS is not covered in the paper. (This paragraph is located at the end of the introduction section.)

Considering the failure mode of the test beam and sufficient shear capacity based on the allowable stress method, the existing short-span simply supported bridges will not experience oblique section failure. So the shear strengthening measures are not covered in the paper. (This paragraph is located at the end of the 4th section.)